# An Interactive Framework for Generating Clinical Data with Human Feedback

Yu Yang*
*Dept. of Electrical & Computer Engineering*
*Duke University*
Durham, USA
yu.yang@duke.edu

Jiafeng Song*
*Dept. of Biomedical Engineering*
*Duke University*
Durham, USA
stuart.song@duke.edu

Zhishuai Liu
*Dept. of Biostatistics & Bioinformatics*
*Duke University*
Durham, USA
zhishuai.liu@duke.edu

Henry Foote
*Dept. of Pediatrics*
*Duke University*
Durham, USA
henry.foote@duke.edu

Rishikesan Kamaleswaran
*Dept. of Surgery*
*Duke University*
Durham, USA
r.kamaleswaran@duke.edu

Pan Xu
*Dept. of Biostatistics & Bioinformatics*
*Duke University*
Durham, USA
pan.xu@duke.edu

*Abstract*—**Evaluating machine-learning models in critical-care settings is particularly challenging. The lack of sufficient data for rare but clinically important cases can lead to unreliable model performance. Clinicians often require specific patient scenarios to assess the robustness of machine learning methods. However, it is difficult to manually construct such patient profiles. Generating patient data for some conditions presents a promising alternative. Therefore, conditional generation methods are needed to create realistic synthetic data that aligns with clinician-defined criteria. To address this challenge, we introduce a novel interactive generative framework that allows clinicians to specify desired patient characteristics and generate synthetic data accordingly. In this paper, we focus on the problem of generating synthetic data for electronic health records (EHR), especially for patients on mechanical ventilation and ECMO where the data is limited. We propose a novel interactive tool InterGenEHR that leverages the generative model with arbitrary conditioning to generate synthetic data conditioned on clinician-specified features. We evaluate our proposed interactive framework using numerical metrics of synthetic data quality and clinically meaningful assessments based on clinician feedback. We also provide a web application that allows clinicians to interactively generate synthetic data based on their requirements and evaluate via clinicians. In summary, we provide an effective tool for validating machine learning methods using clinician feedback tailored to individual patient scenarios.**
**Project Website: https://panxulab.github.io/InterGenEHR**

*Index Terms*—**Generative models, human in the loop, electronic health record, mechanical ventilation and ECMO**

## I. INTRODUCTION

Machine learning (ML) models are increasingly being deployed in critical-care settings, where they are expected to support clinical decision-making [10, 16]. A key enabler of these models is the widespread availability of electronic health record (EHR) data, which captures rich, time-resolved information about patient physiology, interventions, and outcomes. EHRs have been used extensively to develop ML models for outcome prediction, early warning systems, and treatment

recommendations [2, 11]. However, despite their promise, the evaluation and deployment of these models remain difficult due to challenges such as data heterogeneity, distributional shifts across institutions, and the lack of prospective validation. It is crucial to ensure that these models can generalize well to a wide range of patient conditions, especially in high-stakes environments like the ICU.

The training and evaluation datasets often fail to represent the full range of clinical scenarios that clinicians encounter, which makes it challenging to accurately assess model performance and reliability. This is especially problematic when clinicians seek to evaluate the generalization and robustness of machine learning models on specific patients with rare or high-risk conditions. However, these patient types are often underrepresented or entirely missing from existing EHR datasets. To address this limitation, it is crucial to generate synthetic patient data based on partial clinical features specified by clinicians. This capability would allow for more targeted evaluation of model behavior in clinically meaningful but underrepresented scenarios. A particularly pressing example of this challenge arises in the treatment of critically ill patients supported by mechanical ventilation and venovenous extracorporeal membrane oxygenation (VV-ECMO), where data is especially limited and have high bias. These limitations are especially problematic given the complexity and risk associated with managing such patients. Mechanical ventilation provides essential respiratory support by ensuring adequate oxygenation and carbon dioxide removal when spontaneous breathing is insufficient [1], yet inappropriate ventilator settings can cause ventilator-induced lung injury, increasing morbidity and mortality [6]. VV-ECMO adds another layer of complexity, carrying risks such as hemolysis, major bleeding, and thromboembolism. Its successful use requires careful coordination with ventilator management and precisely timed weaning strategies.

In this study, we leverage EHR data from patients undergoing mechanical ventilation and ECMO support to address

---

*Equal contribution.

these critical limitations. We address this gap by introducing an interactive generative framework that allows clinicians to define desired patient characteristics and receive synthetic data with these constraints. The framework InterGenEHR adapts the generative model with arbitrary conditioning [15] to EHR data by conditioning generation while integrating human feedback to make the synthetic data clinically meaningful.

Our interactive tool conditions generation on clinician-specified features such as PaO$_2$/FiO$_2$ ratio, ventilator compliance trajectory, or ECMO sweep gas settings. To realize arbitrary conditioning, we use the Posterior Matching Variational Autoencoder (PMVAE), which learns and performs a mapping from partially observed clinical states to the VAE latent space, ensuring the synthetic data reflects the specified constraints. By incorporating clinician feedback into the generative process, the framework produces synthetic data that is both realistic and tailored to specific scenarios, supporting targeted clinical research and decision making.

We evaluate our approach on a dataset of patients receiving mechanical ventilation and ECMO support. The framework is assessed with quantitative metrics for data fidelity, clinically meaningful scenarios, and an interactive web application that allows clinicians to generate patient-specific profiles. The main contributions of this work are as follows:

- We introduce a novel interactive generative framework InterGenEHR that allows clinicians to specify desired patient characteristics and generate synthetic data accordingly. Our proposed tool can adapt to various arbitrary conditioning generative models and provides a user-friendly web application that allows clinicians to interactively generate synthetic data based on their requirements for designing patient scenarios. We discuss our interactive tool in Section II.
- We validate our proposed tool on a dataset of patients receiving mechanical ventilation and ECMO support. We evaluate the fidelity of the generated data, which shows that we can achieve competitive generation quality compared with more advanced traditional generative models. We also demonstrate the ability of our framework given the clinician's desired features. We discuss the details in Section IV.
- We also demonstrate the effectiveness of our approach in clinically meaningful scenarios. We validate the physics-based constraints in the dataset and show that we can directly integrate these constraints via our tool. We also show that our framework can effectively incorporate clinical feedback and be used in various clinical scenarios. We provide the detailed results and analysis in Section V.

## II. METHODS

### A. Our Framework

As illustrated in Figure 1, our proposed framework Inter-GenEHR leverages a generative model with arbitrary conditioning to create synthetic clinical data within an interactive web application designed for clinician engagement. The process begins with a clinical dataset that is used to train the generative model, enabling it to produce realistic synthetic data based on the input features. This synthetic data is then made accessible through the web application, where clinicians can review, validate, and provide feedback. The feedback collected from clinicians via the web app is systematically fed back into the generative model, allowing for the synthetic data with the clinician-given input features.

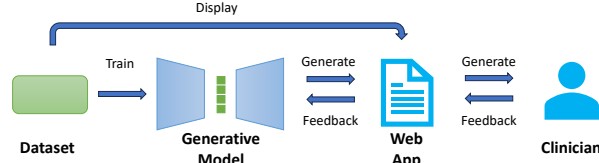

Fig. 1. Overview of our framework InterGenEHR. The clinical dataset trains the generative model to produce synthetic data, which is accessed and validated by clinicians through a web application. Clinician feedback is then used during inference via the generative model, enabling it to generate synthetic data based on clinically meaningful input features.

### B. Generative Models Used in InterGenEHR

In this section, we discuss what type of generative models is suitable for our InterGenEHR framework. In particular, we first introduce the concept of arbitrary conditioning and then use PMVAE as an example to combine with our framework.

*1) Arbitrary Conditioning:* In this section, we discuss the concept of arbitrary conditioning in the context of our proposed clinical setting. In the InterGenEHR framework, clinicians specify a subset of variables and ask the model to infer the remaining features. This is not a standard conditional generation problem, where the model is trained to generate data conditioned on a fixed set of variables. Instead, the conditioning set can vary widely based on clinician input, requiring a more flexible approach. This requires the generative model to be able to generate the data based on arbitrary conditioning sets. We follow [15] to formulate arbitrary conditioning in our clinical use case.

An arbitrary conditioning generative model can model the conditional dependencies between the input features. Several works [3, 14, 15] explored such models, which can be fully leveraged in our framework. In the context of our clinical use case, we denote the patient data samples as $x$ having $d$ features, which represent the health states of the patients. We denote the observed features, which are specified by the clinicians, as $o \subset \{1, \ldots, d\}$. The remaining unobserved features that need to be generated are denoted as $u \subset \{1, \ldots, d\}$. We aim to leverage arbitrary conditional generative models to model $p(\mathbf{x}_u | \mathbf{x}_o)$ for all possible subsets of observed features from the clinicians and unobserved features that need to be imputed.

*2) PMVAE:* We use PMVAE [15] as a representative example to be incorporated to our InterGenEHR framework.

*a) Variational Autoencoder:* VAEs [7] are a foundational class of generative models that enable the modeling of complex data distributions through a structured latent space. We have the following formula in data likelihood $p(\mathbf{x}) = \int p(\mathbf{x}|\mathbf{z})p(\mathbf{z})\,d\mathbf{z}$, where $\mathbf{z}$ represents a latent variable with a lower dimensionality than the observed data $\mathbf{x}$, and $p(\mathbf{z})$ is the

prior distribution over the latent variables. In our setting, $\mathbf{x}$ aggregates heterogeneous variables such as the vital signs and laboratory results, which represent the patients' health states.

The training of VAEs involves optimizing the marginal log-likelihood $\log p(\mathbf{x})$ which is intractable due to the integral over $\mathbf{z}$. To address this, VAEs employ variational inference by introducing an approximate posterior $q_\psi(\mathbf{z}|\mathbf{x})$, known as the encoder, and maximize the evidence lower bound (ELBO) of the data likelihood: $\log p(\mathbf{x}) \geq \mathbb{E}_{\mathbf{z} \sim q_\psi(\cdot|\mathbf{x})}[\log p_\phi(\mathbf{x}|\mathbf{z})] - \mathrm{KL}(q_\psi(\mathbf{z}|\mathbf{x}) \parallel p(\mathbf{z}))$, where $p_\phi(\mathbf{x}|\mathbf{z})$ represents the decoder, modeling the conditional likelihood of the data given latent variables. Both the encoder $q_\psi(\mathbf{z}|\mathbf{x})$ and decoder $p_\phi(\mathbf{x}|\mathbf{z})$ are typically implemented as neural networks.

*b) Posterior Matching:* Posterior Matching [15] is a flexible framework that enhances VAEs for arbitrary conditional distribution estimation, modeling the conditional distribution $p(\mathbf{x}_u|\mathbf{x}_o)$ for non-overlapping subsets of observed features $\mathbf{x}_o$ and unobserved features $\mathbf{x}_u$, where $o, u \subset \{1, \ldots, d\}$. By approximating the partially observed posterior $p(\mathbf{z}|\mathbf{x}_o)$ in the VAE's latent space, Posterior Matching enables conditional inference without modifying the underlying VAE architecture.

The true partially observed posterior is defined as: $p(\mathbf{z}|\mathbf{x}_o) = \mathbb{E}_{\mathbf{x}_u \sim p(\cdot|\mathbf{x}_o)}[q_\psi(\mathbf{z}|\mathbf{x}_o, \mathbf{x}_u)]$, where $q_\psi(\mathbf{z}|\mathbf{x}_o, \mathbf{x}_u) = q_\psi(\mathbf{z}|\mathbf{x})$ is the VAE's encoder with parameters $\psi$. Since computing this expectation is intractable due to the unknown $p(\mathbf{x}_u|\mathbf{x}_o)$, a neural network approximates the posterior as $q_\theta(\mathbf{z}|\mathbf{x}_o)$, parameterized by $\theta$. The training objective minimizes the expected negative log-likelihood: $\mathbb{E}_{\mathbf{x}_u \sim p(\cdot|\mathbf{x}_o)}[\mathbb{E}_{\mathbf{z} \sim q_\psi(\cdot|\mathbf{x}_o, \mathbf{x}_u)}[-\log q_\theta(\mathbf{z}|\mathbf{x}_o)]]$. In practice, the expectation is approximated using a single sample $\mathbf{x}_o$ from the training data and a single sample $\mathbf{z} \sim q_\psi(\cdot|\mathbf{x})$, yielding the loss: $\mathcal{L}_{\mathrm{PM}}(\mathbf{x}, o, \theta, \psi) = -\mathbb{E}_{\mathbf{z} \sim q_\psi(\cdot|\mathbf{x})}[\log q_\theta(\mathbf{z}|\mathbf{x}_o)]$, where $o$ represents randomly sampled observed feature indices. The input $\mathbf{x}_o$ is typically a full data vector $\mathbf{x}$ with unobserved features masked.

In our implementation, we train the VAE encoder $\psi$ and the posterior matching encoder $\theta$ concurrently. During training, we optimize a combined objective consisting of the ELBO and the posterior matching loss $\mathcal{L}_{\mathrm{PM}}$.

## C. Interactive Tool

In this section, we introduce the interactive tool illustrated in Figure 2. This tool is a web application designed to allow clinicians to generate synthetic EHR data interactively based on input features they specify. Users begin by selecting a unique Clinical Sequence Number (CSN), which identifies an individual patient, and a corresponding time point representing a single observation within that patient's clinical timeline.

The interface then presents a modifiable set of physiological features associated with the selected time point. Users can choose a feature (e.g., `bicarb (HCO3)`), adjust its value via a numeric input box or slider, and submit the change. Upon modification, the tool leverages the generative model with arbitrary conditioning to reconstruct the complete clinical state conditioned on the altered input.

The resulting output is visualized using side-by-side bar charts under the "Reconstructed Output" section. Each chart compares the original input values (i.e., the patient's observed data at the selected time point) with the reconstructed values inferred by the model. This allows clinicians to visually assess how the change in one variable may influence related physiological indicators, such as pulse and temperature.

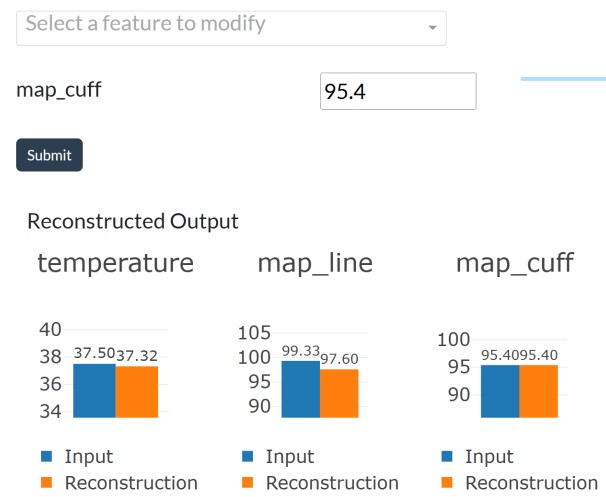

Fig. 2. Interactive Tool. Clinicians select a CSN and time point, then modify physiological states with the tool displaying updated output values and comparative bar plots of original versus modified states to support clinical decision-making.

## D. Extension to Other Scenarios

We currently focus on the VV-ECMO setting and demonstrate the tool on a VV-ECMO cohort using PMVAE. Importantly, the framework itself is model-agnostic and can accommodate a variety of arbitrary-conditioning generative models as well as different clinical scenarios. For the arbitrary conditioning component, alternative backbones from the literature [3, 14, 15] can be substituted. As an illustration of this flexibility, we also report results using Variational Autoencoder with Arbitrary Conditioning (VAEAC) [3]. Extending our framework to new cohorts is straightforward. For instance, our framework can be adapted to the sepsis patient samples. Clinicians follow the same procedure as the VV-ECOM to design and generate the special cases in sepsis scenarios. Likewise, the method generalizes naturally to other high-dimensional EHR settings, where generative models have been shown to scale to hundreds or even thousands of input features by capturing complex dependencies.

## III. EXPERIMENTS DESIGN

### A. Dataset

We collected data from 184 adult patients who underwent venovenous extracorporeal membrane oxygenation (VV-ECMO) at Emory University Hospital between 2015 and 2021. The dataset includes patients with documented VV-ECMO cannulation, including recorded initiation and removal dates, as well as hospital outcomes (in-hospital mortality or discharge

status). We selected patients with vital signs, laboratory values, and ventilation settings recorded at 4-hour intervals, resulting in 184 patients with 15,191 total timesteps. The dataset comprises 41 features representing the patient's physiological state, categorized into four types: vital signs, ventilation settings/measures, laboratory tests, and sedation scores. Following the procedure outlined by [12], each patient's data were segmented into non-overlapping 4-hour intervals. Within each interval, physiological measurements and laboratory variables were aggregated by calculating the median value, thereby minimizing the influence of transient fluctuations and measurement noise. To address missing values, we initially employed forward-filling to propagate the most recent valid measurement forward in time for each patient. Subsequently, any residual missing data were imputed using the population median for the respective variables. This comprehensive imputation method ensured the completeness and consistency of the dataset for further analysis.

### B. Baselines

Our proposed framework targets to generate patient synthetic data based on the clinician's input features. Due to the uniqueness of this application, existing generative models are not explicitly designed for this use case. To evaluate our framework fairly, we designed a comparative study using general-purpose generative models commonly used for structured, tabular time-series data. We use the PMVAE[15] and VAEAC[3] as the generative model in our porposed framework. We denote them as the InterGenEHR-PMVAE and InterGenEHR-VAEAC respectively. We compare our framework against several representative generative models for tabular data, including VAE-based, GAN-based, and diffusion-based approaches.

We include the vanilla VAE [7] as a fundamental latent-variable model that optimizes a likelihood-based objective with a simple prior, serving as a baseline for evaluating the benefits of our InterGenEHR. CTGAN[17] represents the GAN-based methods, offering strong performance on tabular data via its conditional generator and training mechanisms tailored to mixed data types. TabDDPM[9] is included as a recent diffusion-based model that applies noise-based forward–reverse training to learn complex data distributions. TabSyn[4] is a VAE-Diffusion hybrid generative model which provides advanced tabular synthetic data.

This diverse set of baselines allows us to assess the effectiveness of our proposed framework in comparison to well-established and state-of-the-art generative paradigms. All models are trained and evaluated under the same experimental settings, ensuring a fair and consistent comparison.

## IV. VALIDATION ON SYNTHETIC DATA

In this section, we evaluate our proposed interactive tool. The evaluation focuses on three primary objectives: (1) assessing the fidelity of the synthetic data generated by the tool, (2) generating synthetic data from perturbed datasets compared to traditional generative models, and (3) evaluating its data imputation capabilities. We demonstrate that our proposed

interactive tool achieves fidelity comparable to synthetic data produced by traditional generative models and showcases its performance in data imputation and perturbation handling to highlight its capability for arbitrary conditioning.

### A. Evaluation on Synthetic Quality

We evaluated the quality of synthetic data generated by baselines with Maximum Mean Discrepancy (MMD), Root Mean Square Percentage Error (RMSPE), and Mean Absolute Percentage Error (MAPE). These metrics assess the similarity between synthetic and real datasets by examining statistical distributions and feature correlations.

The evaluation of our proposed framework compared to the following generative models: VAE [7], CTGAN [17], TabDDPM [9] and TabSyn[18]. We split our dataset into the train/validation/test sets with 80%, 10% and 10% respectively. We conducted the analysis on a dataset with 41 variables denoted as $K$ variables, comprising 1000 synthetic samples denoted $M$ and 1115 real samples from the test set denoted as $N$. The metrics were computed as follows. All the synthetic data is generated from noise. We would like to compare the synthetic quality of these methods.

For each variable $k = 1, \ldots, K$, we calculated the mean values of the synthetic and real datasets and denoted as $\hat{\mu}^{(k)}\text{syn} = 1/M \sum_{i=1}^{M} x_{i,\text{syn}}^{(k)}$ and $\hat{\mu}^{(k)}\text{real} = 1/N \sum_{i=1}^{N} x_{i,\text{real}}^{(k)}$. The evaluation metrics were then defined as: $\text{MMD} = \max_{1 \leq k \leq K} |\hat{\mu}_{\text{syn}}^{(k)} - \hat{\mu}_{\text{real}}^{(k)}|$, $\text{RMSPE} = 100\sqrt{1/K \sum_{k=1}^{K} \left[(\hat{\mu}_{\text{syn}}^{(k)} - \hat{\mu}_{\text{real}}^{(k)})/\hat{\mu}_{\text{real}}^{(k)}\right]^2}$ and $\text{MAPE} = 100/K \sum_{k=1}^{K} |(\hat{\mu}_{\text{syn}}^{(k)} - \hat{\mu}_{\text{real}}^{(k)})/\hat{\mu}_{\text{real}}^{(k)}|$, where the $M$ is the number of training samples and the $N$ is the number of synthetic samples, $K$ is the number of the features. Results are summarized in Table I, highlighting the performance of each method in generating synthetic data that closely resembles the real clinical dataset. In Table I, it shows that our proposed framework InterGenEHR-PMVAE and InterGenEHR-VAEAC can achieve a similar performance compared with an advanced generative model in generating quality for synthetic data generated from the noise.

TABLE I
FIDELITY OF SYNTHETIC DATA

| | MMD ↓ | RMSPE ↓ | MAPE ↓ |
|---|---|---|---|
| InterGenEHR-PMVAE | 0.44 | 10.07 | 5.93 |
| InterGenEHR-VAEAC | 1.01 | 25.23 | 17.23 |
| VAE | 0.41 | 8.57 | 4.02 |
| CTGAN | 0.67 | 20.76 | 13.49 |
| TabDDPM | 0.32 | 10.58 | 5.15 |
| TabSyn | 0.15 | 8.78 | 4.60 |

### B. Evaluation on the perturbed dataset.

In this experiment, we assess the robustness and stability of our proposed InterGenEHR-PMVAE and InterGenEHR-VAEAC, in comparison to the baseline method VAE, under perturbations in the clinical dataset. The perturbations in the

training data simulate clinician feedback by modifying the features of health states based on existing patient data.

Using our interactive tool described in Section II-C, clinicians can select specific patients (via CSN) and timesteps, then perturb the data to achieve a targeted health state. For instance, a clinician might adjust a patient's mean arterial pressure (e.g., map_cuff) to simulate a hypertensive crisis. Unlike traditional generative models, which struggle to incorporate such perturbations as they do not explicitly model conditional dependencies, our framework leverages arbitrary conditioning to capture these perturbed features as observed inputs, enabling more accurate synthetic data generation. The quality of synthetic samples is quantified using a statistical distance measure, including the MMD, RMSPE, and MAPE.

The results in Table II show that InterGenEHR-PMVAE can achieve competitive results compared with the VAE baseline and InterGenEHR-VAEAC is better than the VAE in MMD. These findings demonstrate that our tools, InterGenEHR-PMVAE and InterGenEHR-VAEAC, enable clinicians to design and explore specific patient scenarios effectively, supporting clinical research and decision-making by generating realistic synthetic data under controlled perturbations.

TABLE II
DATA PERTURBATION

|  | MMD ↓ | RMSPE ↓ | MAPE ↓ |
|---|---|---|---|
| InterGenEHR-PMVAE | 0.37 | 3.91 | 2.11 |
| InterGenEHR-VAEAC | 0.10 | 3.76 | 2.34 |
| VAE | 0.33 | 3.92 | 2.21 |

*C. Evaluation on Data Imputation*

We further evaluate the effectiveness of our approach on a downstream task–data imputation. In this task, the framework generates synthetic patient data conditioned on observed features $x_o$, aiming to reconstruct the missing features $x_u$. Effective imputation indicates the model's ability to generate reliable data given partial observations.

In this experiment, we follow the same data splitting procedure as described in Section IV-A. We train each generative model using the training subset. Then, we simulate partially observed features, representing features specified by clinicians, by randomly masking clinical features in the test subset, and then generate the remaining unobserved features by the trained generative model. We set the mask ratio to be 50%.

Each model attempts to reconstruct the missing features conditioned on the observed data. We quantify reconstruction performance using the mean squared error (MSE) and negative log likelihood (NLL), which measures the discrepancy between the generated values and the true values in the test set. Lower MSE and NLL indicate better imputation quality.

Results are summarized in Table III, clearly demonstrating that our InterGenEHR-PMVAE and InterGenEHR-VAEAC achieve significantly lower MSE compared to the baseline method VAE. And InterGenEHR-VAEAC has the lowest neagtive log likelihood among these methods. We conclude that

the posterior matching, which explicitly models the conditional distribution during training, substantially improves the accuracy and reliability of imputed features. Such improvements can notably enhance the quality of downstream clinical decision-making and predictive modeling tasks, as we will show in the next section, affirming the practical utility of our proposed framework.

| TABLE III DATA IMPUTATION | | | TABLE IV EVALUATION ON SUBSET | | |
|---|---|---|---|---|---|
|  | MSE ↓ | NLL ↓ |  | MSE ↓ | NLL ↓ |
| InterGenEHR-PMVAE | 0.22 | 0.91 | InterGenEHR-PMVAE | 0.20 | 1.26 |
| InterGenEHR-VAEAC | 0.23 | 0.12 | InterGenEHR-VAEAC | 0.52 | 0.51 |
| VAE | 0.94 | 1.76 | VAE | 0.90 | 1.79 |

## V. VALIDATION ON CLINICALLY MEANINGFUL SCENARIOS

In this section, we evaluate our proposed interactive tool's ability to generate synthetic data with high clinical fidelity. The evaluation focuses on three primary objectives: (1) assessing the tool's ability to generate synthetic data that adheres to clinical constraints, (2) evaluating its performance of synthetic data generation conditioned on a curated subset of features of clinical interest, and (3) determining the utility of the generated synthetic data in a downstream machine learning task–mortality prediction. We demonstrate that our proposed interactive tool produces high fidelity synthetic data that satisfies clinical constraints, conditioned on the subset of critical features, and ensures reliable downstream task performance.

*A. Evaluation of Clinically-Constrained Data Generation*

In this section, we evaluate whether the proposed interactive tool can generate clinically meaningful synthetic data. Traditional generative models typically produce synthetic data without incorporating constraints from clinicians, which often results in data lacking clinical relevance. Another challenge is that certain clinical constraints, for example, the physiological equation for pH, are expected to be captured by the generative model when learning the data distribution.

However, such constraints may not hold in the synthetic data due to noise induced by data preprocessing and inadequate learning of generative models. To address the above issue, our proposed framework allows clinicians to directly specify constraints, forcing the constraints to be held in the generated synthetic data. To demonstrate this advantage, we compare our framework with traditional generative models by evaluating their ability to produce clinically valid synthetic data. The evaluation metric, mean absolute error (MAE), measures how well the generated features satisfy those prior clinical constraints, which are essential to ensure clinical plausibility.

Based on previous studies in the literature, we identify the following three constraints. The first equation is the Henderson–Hasselbalch equation [8]:

$$pH = 6.1 + \log\left(hco3/(0.03 \times pco2)\right). \tag{1}$$

TABLE V
MAE FOR CLINICAL CONSTRAINTS

|  | pH MAE↓ | Base Excess MAE ↓ | spo2 MAE ↓ |
|---|---|---|---|
| VAE | 0.04 | 4.09 | 3.15 |
| CTGAN | 0.10 | 5.88 | 3.56 |
| TabDDPM | 0.03 | 4.18 | 2.15 |
| TabSyn | 0.03 | 4.03 | 2.27 |
| InterGenEHR-PMVAE | 0.00 | 0.00 | 0.00 |
| InterGenEHR-VAEAC | 0.00 | 0.00 | 0.00 |
| Real Data | 0.04 | 4.54 | 2.63 |

Secondly, [5] showed that another equation holds:

$$base\_excess = 0.9287 \times hco3 + 13.77 \times pH - 124.58. \quad (2)$$

Finally, we also have the Severinghaus equation, showing $O_2$ is binding to hemoglobin [13]:

$$spo2 = \left(23400/(pao2^3 + 150 \times pao2) + 1\right)^{-1}. \quad (3)$$

We calculate the MSE to evaluation the deviation of the generated value from the value calculated by the above equations to assess how baseline methods obey these constraints.

Table V shows that traditional generative models frequently violate these constraints, producing unrealistic samples. In this task, we observe that even real data does not strictly follow the constraints, and traditional generative models also exhibit deviations. This is because noise is introduced during data preprocessing, which contributes to these violations. Such violations can lead to inaccurate predictions in downstream tasks, highlighting the importance of constraint-aware data generation in clinical applications. Our proposed method can achieve exactly 0 MSE, because these constraints are forced to be held when specifying the conditions. We also show in Table VI that our framework InterGenEHR can generate synthetic data with comparable level of deviation to the real data when only one of those constraints is forced to be held.

TABLE VI
MAE FOR CLINICAL CONSTRAINTS

|  | pH MAE ↓ | Base Excess MAE ↓ | spo2 MAE ↓ |
|---|---|---|---|
| InterGenEHR-PMVAE_pH | 0 | 4.56 | 2.60 |
| InterGenEHR-PMVAE_Base_Excess | 0.04 | 0 | 2.61 |
| InterGenEHR-PMVAE_spo2 | 0.04 | 4.01 | 0 |
| InterGenEHR-VAEAC_pH | 0.00 | 4.02 | 2.60 |
| InterGenEHR-VAEAC_Base_Excess | 0.01 | 0.00 | 2.59 |
| InterGenEHR-VAEAC_spo2 | 0.01 | 4.03 | 0.00 |

### B. Subset for Evaluation: Clinically Salient Features

In this section, we evaluate the model's performance on a subset of clinically important features identified through consultation with domain experts. Specifically, we worked with clinicians to select 14 key variables that are considered critical for the management of VV-ECMO patients. These features serve as either conditioning inputs or targets to be generated by the model, depending on the experimental setup. Focusing on this subset allows us to simulate scenarios where clinicians are particularly interested in the model's ability to

accurately handle or generate clinically relevant data. The selected features are grouped into three primary categories.

(1) **Oxygenation-related features** are crucial indicators of a patient's oxygenation status: `pf_sp`, `spo2`, `pao2`, and `sao2`. (2) **Ventilation-related features** provide insights on respiratory mechanics and acid-base balance: `base_excess`, `paco2`, `hco3`, and `ph`. (3) **Other markers of interest (including Hemolysis)** cover various laboratory values indicating other physiological processes, such as hemolysis or organ function. These features are: `hemoglobin`, `platelets`, `creatinine`, `lactate_dehydrogenase` and `bilirubin_total`.

Then, we evaluate our proposed tool and compare it with VAE on the data imputation in this subset. We follow the same settings in Section IV-C, with the above-mentioned features unmasked and remaining features masked. We summarized the results in Table IV, which shows that InterGenEHR-PMVAE and InterGenEHR-VAEAC outperform VAE on generating high-fidelity synthetic data. InterGenEHR-PMVAE performs well in MSE and is not as good as InterGenEHR-VAEAC in NLL. It validates that our tool is capable of generating clinically meaningful data when conditioned on the subset of features identified as important by clinicians.

### C. Utility-Downstream Task: Mortality Prediction

In this section, we evaluate the utility of the generated synthetic data by assessing its performance in a downstream task, predicting in-hospital mortality.

For our VV-ECMO dataset, we have time series data for each patient and a corresponding in-hospital mortality outcome. We preprocess the dataset by selecting the 24 hours prior to the final recorded timestep, which corresponds to the last six timesteps of patient data. These six timesteps are concatenated into a single feature vector. In this setting, we use 145 patients for training, 18 for validation, and 18 for testing. With 41 features per timestep, the concatenated patient vector contains 246 features in total. During training, we include the mortality label as an additional feature. We train the generative model on the processed data and generate synthetic samples to evaluate performance on downstream tasks.

After we get the synthetic data from the generative model, we employ the "train on synthetic, test on real" (TSTR) paradigm to evaluate the quality of the synthetic data generated by VAE and our InterGenEHR-PMVAE and InterGenEHR-VAEAC. A mortality classifier was trained on the synthetic datasets by an MLP and evaluated on the real VV-ECMO dataset. We also train a mortality classifier on the real data as the base metric to evaluate these generative models.

We evaluated model performance using the Area Under the Receiver Operating Characteristic Curve (AUROC) and accuracy, both are commonly used metrics for assessing binary classification models. To quantify the discrepancy, we calculated the difference between the AUROC score of a classifier trained on real data and that trained on synthetic data, with a smaller difference indicating higher utility of the synthetic data. We also demonstrate the accuracy to show the

performance. For our proposed framework, we generate the synthetic data based on random masks.

Results are presented in Table VII, where differences in AUROC values reflect the effectiveness of each method. Our proposed InterGenEHR-PMVAE and InterGenEHR-VAEAC, conditioned on the given observed features, demonstrates smaller discrepancies and higher accuracy than the VAE model. InterGenEHR-VAEAC outperforms other methods in AUROC. This suggests that our model can generate clinically meaningful synthetic data when conditioned on clinically relevant features identified by clinicians.

TABLE VII
UTILITY EVALUATION (REAL-SYNTHETIC)

| Method | AUROC ↑ | Difference ↓ | Accuracy↑ |
|---|---|---|---|
| InterGenEHR-PMVAE | 0.78 | 0.17 | 0.89 |
| InterGenEHR-VAEAC | 0.86 | 0.09 | 0.89 |
| VAE | 0.77 | 0.19 | 0.83 |
| Real Data | 0.95 | 0.00 | 0.89 |

## VI. CONCLUSION

In this paper, we propose a novel interactive generative framework that enables clinicians to specify desired patient characteristics for synthetic data generation. Our approach is model-agnostic and adaptable to various conditioning generative models, demonstrated using InterGenEHR-PMVAE and InterGenEHR-VAEAC. The accompanying user-friendly web application allows clinicians to intuitively interact with the system and generate synthetic patient data tailored to specific clinical needs. We validated the framework using a dataset of patients receiving mechanical ventilation and ECMO support. Our results show that the generated data maintains high fidelity and quality. Finally, we demonstrated the framework's utility in clinically meaningful use cases. This work bridges the gap between advanced generative models and clinical practice.

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
