# OpenReview forum: "An Interactive Framework for Generating Clinical Data with Human Feedback"
_IEEE.org/EMBS/BHI/2025/Conference — BHI 2025_

### Official Review · Reviewer_kXq9 · 2025-07-16
**An Interactive Framework for Generating Clinical Data with Human Feedback**

**Confidence:** 3
**Clarity Of Writing:** good
**Clinical Significance:** great
**Methodological Novelty:** good
**Overall Rating:** 7

**Experiments And Results:**

great

**Questions For The Authors:**

How would your framework handle scenarios with thousands of features or multiple modalities (e.g., free text, images)?

Does enforcing constraints on one feature degrade the realism of others? Is there a way to quantify such trade-offs?

**Strengths:**

This work targets an urgent and understudied problem, generating data for rare, high-stakes ICU scenarios (ECMO, ventilation). The inclusion of a web interface for real-time clinician interaction is highly relevant and increases translational potential. The PMVAE-based framework supports flexible conditioning on any subset of features, making it versatile for clinical experimentation. Also, the explicit modeling of clinical equations (e.g., Henderson-Hasselbalch) and validation of constraint adherence is a very valuable feature.
The evaluation in this work is conprehensive, it includes fidelity, perturbation, imputation, and downstream classification experiments, as well as evaluations on clinically meaningful feature subsets.

**Summary Of The Paper:**

This work proposes an interactive framework for clinician-guided synthetic data generation in critical care, specifically targeting mechanical ventilation and ECMO scenarios, which suffer from data scarcity. The method leverages a Posterior Matching Variational Autoencoder (PMVAE) to support arbitrary conditioning on clinician-specified features. The system includes a web-based interface where clinicians can modify input values and receive full patient profiles reflecting those changes. The approach is evaluated for fidelity, robustness to perturbations, data imputation, and its ability to respect clinical constraints and support downstream tasks like mortality prediction.

**Weaknesses:**

Although PMVAE performs well, its comparisons to baseline models like VAE, CTGAN, and TabDDPM are largely at a surface level. There's no comparison to more advanced conditional or hybrid generative models (e.g., conditional flows, diffusion-VAE hybrids). Metrics such as diversity and overfitting risk (e.g., via memorization or privacy metrics) are also absent.

Posterior matching via learned approximations is data and architecture-sensitive. There is no discussion on potential failure modes, such as unstable conditioning or conflicting feature specifications. The work would also benefit from discussion on performance and usability at larger scale (e.g., thousands of patients or high-dimensional EHR data).

The constraint satisfaction is achieved by hard-coding conditions, which raises questions about whether the model learns the clinical logic or simply conforms to enforced values.

---

### Official Review · Reviewer_bsDx · 2025-07-17
**An Interactive Framework for Generating Clinical Data with Human Feedback**

**Confidence:** 5
**Clarity Of Writing:** good
**Clinical Significance:** good
**Methodological Novelty:** good
**Overall Rating:** 6

**Experiments And Results:**

good

**Questions For The Authors:**

Can you clarify how the tool enforces constraints exactly during generation?
Does it project latent samples to satisfy the equations, or does it train the decoder to strictly obey constraints? Understanding this would clarify whether PMVAE generalizes or simply enforces rules.

How generalizable is the framework to other datasets or conditions?
While ECMO patients are a good test case, would the same method work for sepsis, trauma, or cardiac conditions? Evidence or discussion of portability would raise the impact of the paper.

What does "model-agnostic" mean in practice?
You mention that the tool could be adapted to other conditioning models besides PMVAE, but no such experiments are shown. Could you provide evidence (or limitations) of this claim?

**Strengths:**

1- Introduces a novel, interactive human-in-the-loop tool that enables clinician-guided conditioning of synthetic patient data.

2- Demonstrates effective constraint-based data generation aligned with clinical physiology.

3- Validates the synthetic data on multiple fronts: fidelity, imputation, perturbation handling, and utility in downstream tasks (e.g., mortality prediction).

4- Includes comparisons against strong baselines (VAE, CTGAN, TabDDPM) using comprehensive metrics.

5- Offers a practical web interface that supports real-time interaction and visual feedback for clinicians.

**Summary Of The Paper:**

This paper proposes an interactive framework for generating synthetic electronic health record (EHR) data, particularly for rare and complex critical-care scenarios such as patients on mechanical ventilation or ECMO. The method allows clinicians to specify partial patient characteristics and generate matching synthetic data using a Posterior-Matching Variational Autoencoder (PMVAE).

**Weaknesses:**

Major Comments:

1- The dataset includes only 184 patients, which may not be sufficient for training deep generative models robustly.

2- PMVAE outperforms others in constraint satisfaction because it explicitly hard-codes the constraints during conditioning — this gives it an unfair advantage over unconstrained models in that comparison.

3- Web application functionality is described but not evaluated with user studies (e.g., clinician usability, effectiveness, or adoption).

4- The framework does not incorporate uncertainty quantification or confidence bounds on generated outputs, which is critical for clinical decision-making.

5- Some claims about model generalizability (e.g., "model-agnostic" design) are made but not tested with other model types.

Minor Comments:

1- Typo: “We conclude from Table IV that that our proposed…” (repetition of “that”).

2- Figures 1 and 2 are helpful but could be better labeled and more readable (e.g., font size, color contrast).

---

### Official Review · Reviewer_GiRT · 2025-07-18
**Review of 'An Interactive Framework for Generating Clinical Data with Human Feedback'**

**Confidence:** 3
**Clarity Of Writing:** excellent
**Clinical Significance:** good
**Methodological Novelty:** great
**Overall Rating:** 7

**Experiments And Results:**

excellent

**Questions For The Authors:**

- How scalable is PMVAE to higher-dimensional EHR datasets with hundreds of variables or longer time horizons?

- Did you conduct formal usability testing with clinicians to assess the framework’s practical value?

- How do you plan to ensure privacy-preserving guarantees when generating synthetic data?

- How does the model perform under conflicting or unrealistic constraints specified by users

**Strengths:**

- Clinically Relevant Problem: Addresses data scarcity for rare but critical conditions like VV-ECMO, where real-world data is extremely limited.
- Interactive Framework: Provides a human-in-the-loop system with a clinician-facing web application, supporting real-time constraints and visualization.
- Arbitrary Conditioning: PMVAE allows generation under flexible feature constraints, improving practicality for stress-testing models in edge cases.
- Robust Evaluation: Includes fidelity metrics (MMD, RMSPE, MAPE), constraint adherence, imputation tasks, and downstream predictive utility.
- Physiological Plausibility: Explicitly validates constraint enforcement using established clinical equations (e.g., Henderson–Hasselbalch, Severinghaus).

**Summary Of The Paper:**

This paper introduces an interactive framework for generating synthetic clinical data under clinician-specified constraints, targeting rare ICU scenarios such as mechanical ventilation and VV-ECMO. The system uses Posterior Matching Variational Autoencoder (PMVAE) for arbitrary conditioning and provides a web-based interface for real-time clinician interaction. Synthetic data quality is evaluated using distributional metrics (MMD, RMSPE, MAPE), constraint adherence, imputation performance, and downstream utility in a train-on-synthetic, test-on-real (TSTR) mortality prediction task. Results show PMVAE produces realistic, constraint-compliant synthetic data and outperforms a standard VAE in imputation accuracy and downstream AUROC (0.785 vs. 0.768), approaching real data performance (0.954).

**Weaknesses:**

- Limited Dataset and Generalizability
Evaluation is based on 184 patients from a single center, limiting the ability to generalize findings across institutions and populations.

- Although the framework is designed for clinician interaction, the paper does not report systematic clinician feedback or usability studies beyond technical metrics.

---

### Official Review · Reviewer_Ka7k · 2025-07-18
**An Interactive Framework for Generating Clinical Data with Human Feedback**

**Confidence:** 3
**Clarity Of Writing:** great
**Clinical Significance:** great
**Methodological Novelty:** good
**Overall Rating:** 7

**Experiments And Results:**

good

**Questions For The Authors:**

1. In Table 2 (data perturbation experiment), VAE outperforms PMVAE—could you provide an explanation for this result?
2. Could you elaborate on the individual contributions of MMD, RMSPE, and MAPE to model assessment? Is one more suitable for clinical use cases, and if so, why?

**Strengths:**

1. The introduction and motivation for PMVAE and arbitrary conditional generation are clearly articulated.
2. The paper includes a comprehensive comparison against strong baselines (e.g., VAE, CTGAN, TabDDPM), situating PMVAE well within the generative modeling literature.
3. Evaluation spans multiple dimensions, including robustness to perturbation, and clinical utility, strengthening the validity of the claims.
4. The use of a clinically meaningful downstream task (mortality prediction) showcases the practical utility of the generated data.

**Summary Of The Paper:**

This paper proposes an interactive framework for generating synthetic clinical data with human-in-the-loop feedback from clinicians, focusing on electronic health records (EHR) for patients undergoing mechanical ventilation and ECMO. The proposed model, PMVAE, supports arbitrary conditional generation and incorporates clinical feedback to improve the plausibility and utility of the generated data. The quality of the synthetic data is assessed through both quantitative metrics and clinical validation, including a downstream task: mortality prediction.

**Weaknesses:**

1. The rationale behind the choice of evaluation metrics—MMD, RMSPE, and MAPE—could be better explained. It is unclear what each metric captures uniquely and how they complement one another.
2. Readers may benefit from guidance on which metric is most meaningful in which scenario, especially for practitioners less familiar with synthetic data evaluation.
3. Equations (9)–(11) define constraints, but the implementation details of how these constraints are incorporated into the generation framework are not clearly explained.